# The Effect of Climate Change on Food Crop Production in Lagos State

**DOI:** 10.3390/foods11243987

**Published:** 2022-12-09

**Authors:** Tawakalitu Titilayo Tajudeen, Ayo Omotayo, Fatai Olakunle Ogundele, Leah C. Rathbun

**Affiliations:** 1Department of Forestry and Environmental Resources, North Carolina State University, Raleigh, NC 27695, USA; 2Department of Geography and Planning, Lagos State University, Ojo, Lagos PMB 0001, Nigeria

**Keywords:** climate change, food crops, agriculture, maize, rainfall

## Abstract

Climate change is set to be particularly disruptive in poor agricultural communities. This study examines the effects of, and farmer’s perceptions of, climate change on farming practices for cassava and maize in Lagos, Nigeria. Analysis of weather data from 1998 to 2018 (the most recent available) reveals little impact on cassava yield but a significant impact on maize yield. Furthermore, survey results indicate that farmers in this area are currently implementing techniques to adapt to changes in climate based on the type of crop grown. Agriculture in Lagos, Nigeria, is largely rain-fed and climate change negatively impacts crop productivity by decreasing crop yield and soil fertility, limiting the availability of soil water, increasing soil erosion, and contributing to the spread of pests. A decline in crop production due to climate change may be further exasperated by a lack of access to farming technology that reduces over-reliance on the rain-fed farming system and subsistence agriculture. This study indicates that there is a need for initiatives to motivate young and older farmers through access to credits, irrigation facilities, and innovative climate change adaptive strategies.

## 1. Introduction

Climate change is a major global concern that is greatly reshaping the environment and continuously altering Earth’s ecosystems [1]. The pace of climate change has increased in recent years when compared to the last century [2,3,4,5]. Since the nineteenth century, there has been a global increase in average temperature by 0.9 °C [1]. Projections indicate warming will continue with an average increase of 3–4 °C possible over the next century [6]. As the Earth warms generally, average temperatures rise throughout the year, but the increases may be more significant in certain seasons than in others [7]. Climate change leads to a distortion of seasonal patterns and consequently, changes in the pattern of rainfall and temperature [8].

Climate change contributes to decreased food production, which has become more pronounced over the last two to three decades [6]. Though this is a global problem, it is more prevalent in developing countries, especially sub-Saharan Africa which is among the most affected in the tropical world [6,9]. The varieties of food crops cultivated in these regions are heavily impacted by changes in climate. Gray [10] reported that increases in regional temperatures as a result of climate change, particularly in the tropics, can lead to heat stress for all types of crops. The majority of countries in sub-Saharan Africa are agrarian, with much of their populations residing in rural areas. These rural communities depend heavily on agriculture as their means of livelihood [9], and farming practices largely rely on direct rainfall rather than irrigation [11].

It is predicted that climate change is having and will continue to have significant negative impacts on crop production in Nigeria, contributing to food security challenges that lead to adverse socio-economic impacts and changes in ecological conditions [6,12]. Generally, rainfall variation is projected to continue to increase in Nigeria [13]. Precipitation in southern Nigeria is expected to rise, exacerbating the risk of flooding and possible submersion of coastal cities [14,15]. Droughts have also become constant in Nigeria and are expected to continue in Northern areas [16], arising from a decline in precipitation and a rise in temperature [17,18]. Hence, the change in climate has already begun to trigger drought and flooding events that adversely affect crop production throughout Nigeria [13]. This onset of seasonal rains and drought has led farmers to adapt planting dates to later in the season in certain ecosystems (e.g., mangrove swamps, rain forests, and parts of the Sahel and Guinea Savanna) [19].

Over 80% of crop production in Nigeria is dependent on rainfall [19]. Water availability has been shown to be influenced by rising temperatures and longer growing seasons. Increasing evaporative demand is predicted to increase crop irrigation by up to twenty percent globally by 2080 and up to fifteen percent in Nigeria [20,21]. Further, the frequency of storms and intense bursts of rainfall may reduce crop yield, or destroy it completely, and contribute to significant soil erosion and excessive flooding, especially in the tropics [18,22]. Uncertainty around anticipated weather can impact decisions around crop production (e.g., date of planting, seed purchasing, date of harvest), leading to food shortages [12]. Farmers in Nigeria have reported that climate change is causing uncertainty in the length and onset of the farming season, longer and shorter periods of rainfall, and reduced harmattan (a dust-laden wind with very little humidity on the Atlantic coast of Africa) [18]. In addition, rainfall and temperature fluctuations are also associated with increases in the severity and occurrence of plant diseases and insect outbreaks, both which can further suppress crop production and make farming more challenging [23].

A large quantity of food consumed in Nigeria is produced through small-scale agriculture (i.e. rural farming). These farmers constitute approximately 80 percent of the country’s farming population [9,24] and have low agricultural productivity [25]. Currently, a wide gap exists between overall food crop production and Nigeria‘s growing population, with food production increasing arithmetically and the population increasing geometrically [6,9]. All of this indicates that climate change may undermine efforts to address existing and future food security in this area [6,12]. There is a need to understand the impacts of climate change on food security in Nigeria through a robust academic framework [26]. Estimating the impacts of climate change on food security is important for the development of policy that can support interventions and allocate resources focused on adaptive farming practices to ensure secure food production [27,28].

FAS Lagos estimated cassava and maize to be the highest farming crops within Nigeria in 2020 [29]. This study examines the impact of temperature and precipitation on cassava and maize production within Lagos, Nigeria. The study analyzes these trends through the context of cassava and maize production over a period of 30 years to determine which climate variables are impacting crop production in order to assess what future production might look like. In addition, this study assesses the perception of farmers in Lagos of the effects of climate change on food crop production and their coping strategies to date. Farmers in this area are cursorily aware of the term climate change and are currently implementing mitigation practices. This study builds on the body of existing research in Nigeria and contributes additional information for Lagos (an area not previously studied) that can be used to inform policy- and decision-makers about climate change-induced agricultural productivity loss.

## 2. Materials and Methods

### 2.1. Study Area

The state of Lagos, Nigeria, lies between latitude 6° and 7° N and longitude 2° and 5° E [30] and is bounded on the north and east by Ogun State; in the west, it shares borders with the Benin Republic, and its south opens into the Atlantic Ocean (Figure 1). Lagos is 356,861 hectares in size. Just under half of the hectares in Lagos (47% or 169,613 hectares) are designated as agricultural; of which only 30% is currently being utilized as such [31]. Lagos State contains very little arable land [32], and the city of Lagos has a tropical climate. The average annual temperature in Lagos is 27.0 °C, and the average annual rainfall is 1693 mm [33]. Eze et al. [34] proposed that the low-lying nature of Nigeria’s 800 km coastline, from Lagos to Calabar, makes the food systems of this region even more vulnerable to climate change due to seawater flooding within the coastal freshwater resources negatively impacting the inland fisheries and aquaculture.

The study area for this research includes three farm settlements selected from the Badagry, Ikorodu, and Epe local government areas of Lagos, Nigeria. These sites were selected based on their status as rural and agricultural production settlements within Lagos. Lagos state has a landmass of 356,861 hectares, of which 47% (169,613 hectares) are designated for agriculture, with just 30% of the land currently being utilized as such [31]. The five farm settlements located within the study area are comprised of over 140 registered crop farmers and 5000 farmers engaging in mixed agricultural and food production activities [35]. The indigenes of the Ikorodu division are mostly traders, farmers, and anglers located along the Lagos Lagoon foreshore. The main occupations of the Epe inhabitants include fishing and farming.

### 2.2. Sampling Procedures, Data Collection and Analysis

#### 2.2.1. Climate Analysis

Climate data were obtained from the National Bureau of Statistics DataMart (https://nigerianstat.gov.ng/elibrary (accessed on 23 November 2022)) and Central Bank of Nigerian bulletin (http://statistics.cbn.gov.ng/cbn-onlinestats/DataBrowser.aspx (accessed on 23 November 2022)) and collected by the Nigeria Metrological Agency (NIMET). Temperature and rainfall data came from the Lagos-Ikeja-Marina weather station and was available from 1995 to 2018. Monthly minimum and maximum temperatures were publically available for a subset of the years, and annual minimum and maximum temperatures were available for the remaining. All data were congregated into annual values for the years of interest. Monthly rainfall measurements were available for all years, from which total annual rainfall was calculated. The World Meteorological Organization [36] defines the classical period for averaging climate variables as 30 years. As such, the existing time-series data on rainfall (total annual) and temperature (annual minimum and maximum) across the 30 years was used in the analysis. Time series graphs were developed for the climate variables to assess trends and variability.

Food crop production data was collected by the Lagos Ministry of Agriculture and Co-operative and obtained through the National Bureau of Statistics (https://nigerianstat.gov.ng/elibrary (accessed on 23 November 2022)). The statistical report provides crop yield (tonnes per thousand) as well as hectares of production for cassava and maize within Lagos state from 1995 to 2018. Only the years where crop production data were available in correspondence to climate data were used to develop the linear regression. Time series graphs for cassava and maize production and hectares of production were developed to assess trends and variability.

To assess the relationship between crop production and climate variables, a multiple linear regression model was fit using least squares regression. All possible models were fit using the R software package (version 4.1.2) [37]. The final models were selected using adjusted R square values defined as:(1)$${\mathrm{adjusted}\;R}^{2}=\frac{\left(1-R^{2}\right)\left(N-1\right)}{N-p-1}$$

where:

R^2^ = the coefficient of determination;

N = the sample size;

p = the number of parameters in the model.

#### 2.2.2. Farmer’s Perceptions

A survey questionnaire was used to gather information on farmers’ perception of climate change, its effects on farming, and recently implemented mitigation practices. The survey included Likert scale, open-ended, and multiple-choice questions. The link to a copy of the survey has been attached as Appendix A. Demographic questions included those regarding age, gender, marital status, tribe, income, and education level. Additional questions were included to assess a farmer’s level of awareness of climate change, including patterns of change, perceptions of effects on food production due to changes in climatic, and mitigation strategies implemented throughout recent years.

The Farm Settlement Scheme in Nigeria can be described as a government intervention designed to advance rural development and promote efficient utilization of land resources for farming [38]. This scheme was used to define the sampling area for the study. The farm settlement areas of Epe, Ikorodu, and Badagry within Lagos were selected, as agricultural activities within these settlement areas are the primary source of income for the local residents. A list of farmers was provided by the Farm Settlement Association for each of the following farm settlements:(a)Ajara farm settlements in Badagry;(b)Odongunyon and Imota farm settlements in Ikorodu; and(c)Agbowa farm settlement in Epe.

Famers from each of the three lists were randomly selected to participate in the survey. In each of the farm settlements, sixty farmers (twenty of each who farmed cassava, maize, and leafy vegetables) were randomly selected. Leafy vegetables are those with leaves and succulent young shoots picked for consumption. The leafy vegetable farmers were selected to understand if climate change affects them differently from Cassava and Maize farmers. This resulted in 180 questionnaires in total.

After seeking permission from each of the Farm Settlement Association’s Heads, survey questionnaires were distributed at a monthly association meeting during the months of January and February 2019. Not all surveys were filled out and submitted directly during the same monthly association meeting. For any non-responsive participants, follow-up was done at the next monthly association meeting. In some areas where meetings were not held, the surveys were distributed door-to-door. All participants were notified of the survey questionnaire at least one week prior to receiving it and participation was voluntary.

To assess the comprehensibility of the term climate change, open-ended questions were asked to a subset of respondents. These questions were designed to investigate the linkages between a farmer’s education level and their understanding of scientific terms regarding climate change. In addition, this provided an opportunity for farmers who were interested in the subject of this study to ask their own questions.

Descriptive (frequency, percentage, and trend charts) statistics were used to understand the socio-demographic information of the respondents. Inferential statistics were used to examine the variation in adaptive strategies based on the type of crop grown. One-way ANOVAs were used to assess the variation in adaptive strategies adopted by the farmers and their perception of the pattern of climate change phenomenon based on years of farming experience.

## 3. Results

### 3.1. Climate Analysis

#### 3.1.1. Time Series

The first two ten-year periods (1989 to 1998 and 1999 to 2008) for annual maximum temperature show similar trends, with the most recent ten-year period (2009 to 2018) showing differences in the average values as well as increased variability through time (Figure 2). The average annual temperature throughout all years was 27.4 °C, with a standard deviation of 0.73 °C. The most obvious fluctuations in temperature have occurred within the last ten years (2009–2018), which recorded the lowest (21.1 °C) and highest (34.7 °C) values.

Total annual rainfall in this area shows an oscillating trend through time (Figure 3). The average total rainfall across the thirty-year period was 1542 mm with a standard deviation of 257.64 mm. The two most recent ten-year periods (2009 to 2019 and 1999 to 2008) show similar trends. However, there is considerable fluctuation for the years 1989–1998, which includes both the lowest annual total rainfall (1040 mm in 1998) and the highest total annual rainfall (2020 mm in 1997). A small decrease in rainfall occurred during the last ten years (2009–2018), with the highest and lowest total rainfall for this period being 1761 mm and 1319 mm, respectively. The average annual rainfall across all years was 1517 mm, with a standard deviation of 221 mm.

The trend of cassava yield in Lagos shows a steady rise in output from 72.66 thousand tonnes in 1995 to a high peak of 970 thousand tonnes in 2010 (Figure 4). After this time, a sharp decline occurred during the years 2011 through 2013, after which another sharp rise is seen in 2014 and 2015. The most recent four years indicate a steady yield. This same trend applies to maize, with a more consistent and gradual rise in yield, from 47 thousand tonnes in 1995 to its highest peak in 2010 of 223.06 thousand tonnes. There was also an increased yield of maize in 2014 and a moderate decline through 2018. The average crop yield for cassava was 590 tonnes, with a standard deviation of 293, and for maize was 110 tonnes with a standard deviation of 62. For cassava and maize, only the wet seasonal yield data were available for the years 2011 through 2013, which is the reason that such a sharp drop occurred in the data, not an indication that yields decreased significantly. Removing this data did not influence the result, so it was kept in the analysis.

There was a consistent rise in the cultivated area over time for both Cassava and Maize from 1995 to 2011 and a decline from 2013. The average land size for cassava production is 42 thousand hectares with a standard deviation of 19, while the average land size for maize production is 55 thousand hectares with a standard deviation of 30 (Figure 5). The land area for cassava was 7.654 thousand hectares in 1995, and it increased gradually to 69.67 thousand hectares in 2017. In addition, the land area for maize production increased gradually from 25 thousand hectares in 1995 to 109.62 in 2018. This trend also corresponds to the production output recorded for both crops.

#### 3.1.2. Multiple Linear Regression Models

None of the climate variables showed a strong correlation to cassava yield; the annual minimum temperature had the strongest correlation at 0.39, while the annual maximum temperature and annual total rainfall values were 0.20 and −0.09, respectively. As expected, there was a strong direct correlation between the hectares of production and cassava yield (0.98). The adjusted R^2^ values for all possible linear regression models can be found in Table 1.

Where:

C = Annual Cassava Yield;

T_min_ = Minimum Annual Temperature;

T_max_ = Maximum Annual Temperature;

R = Total Annual Rainfall;

A_c_ = Hectares of Cassava Production.

The selected linear regression model for this analysis had an adjusted R^2^ value of 0.96 and was found to be:C = 664.67 − 28.1 T_min_ + 14.21 A_c_.(2)

Maize yield showed strong negative correlations to climate variables. The annual minimum temperature was more strongly correlated (−0.61) to maize yield than the annual maximum temperature (−0.21). Total annual rainfall showed a little-no significant correlation (0.05) with maize yield. As expected, there is a strong correlation between the hectares of production and maize yield (0.82). The adjusted R^2^ values for all possible linear regression models can be found in Table 2.

Where:

M = Annual Maize Yield;

T_min_ = Minimum Annual Temperature;

T_max_ = Maximum Annual Temperature;

R = Total Annual Rainfall;

A_m_ = Hectares of Maize Production.

The selected linear regression model for this analysis had an adjusted R^2^ value of 0.82 and was found to be:M = 878.62 − 17.07 T_min_ − 0.003 R − 14.35 T_max_ + 1.55A_m_.(3)

### 3.2. Farmers’ Perception Analysis

The majority of respondents were male (76.1%) and over the age of fifty (44.4%). Less than seven percent of the respondents were under the age of thirty (6.7%). No significant differences in gender, or age were seen across the different farm settlement areas. Demographics for the respondents to the survey questionnaire can be found in Table 3.

Significant differences were found for income levels across the different farm settlement areas (Table 4). Badagry had more farmers within the lowest income class, yet the percentage within the highest income classes was not significantly different from the other two farm settlement areas. The farmers in this settlement area also had the most farmers working the smallest areas of land (1 to 5 plots of size 0.165 acres), 70 percent as compared to 62 percent in Epe and 42 percent in Ikorodu.

The majority of respondents received more than a primary education (77 percent); with 31 percent receiving a secondary education and just over 46 percent receiving a tertiary education (Table 5). The majority have been farming for more than fifteen years (73.9%) and self-identified as being aware of climate change (79.4%). In addition, the respondents who were aware of climate change indicated that they were informed through a variety of sources ranging from television, association meetings, and conferences.

Questionnaire responses as well as individual discussions with farmers, indicate that they understand the impacts of climate change on their crop yield. A large majority (81%) of respondents indicated that pest and disease outbreaks appear to be increasing (Figure 6). In addition, this was seen to have a negative impact on crop yield. The majority of respondents also perceive rainfall (73%) and temperature (53%) to be increasing, although those respondents from Epe were twenty percent less likely to perceive rising temperatures compared to the other settlement areas. An ANOVA test (*p*-value = 0.002) indicates that the variation in the perception of increasing temperature is based on years of farming experience and the majority of farmers equate rising temperatures (51.1%) and rainfall (73.9%) with higher crop yield. Harmattan events were also perceived to be decreasing in this area.

Answers to the open-ended questions indicate that those respondents that were un-aware of the term climate change were still concerned about the negative effect that cli-mate change will have on their crop yield. They expressed concern about the issue asso-ciated with the late onset of rain and moderate drought, leading to heavy downpours creating soil clogs and loss of vegetable fields and fertile soils. “I tend not to plant until after two or three downpours because in most cases, the first few rainfalls are often heavy, pulling down the stems, and washing away the seeds, and fertilizer”, expressed one of the respondents. Seven out of nine farmers aged sixty or older stated that they believe the environment is not as hot as it is currently compared to when they started farming. 

Respondents were also asked about the various strategies they use to cope with cli-mate change (Figure 7). The ANOVA highlights that there is a significant variation in the type of strategies adopted. Strategies ranged from manure application (88.3%), fertiliza-tion (77.2%), chemical application (95.6%), and multiple cropping (77.2%), with some strategies such as fertilization, mulching, irrigation, and land expansion varying based upon the type of crop farmed. About 25% of vegetable farmers expressed their concern about complaints from their customers due to the smell of applying chemicals and ferti-lizer in their fields. Some vegetable farmers discussed using manure instead of fertilizer. Approximately 65 percent of respondents considered vegetables highly sensitive to water availability, and there was an indication that they use irrigation to cope with the lack of rainfall. Mulching was the least used coping strategy, and there was an indication that mulching has a high labor cost. Lastly, more than 75 percent of the respondents adopt multiple cropping. Respondents indicated that multiple cropping meets both land man-agement and economic objectives. One of the respondents indicated, “I have limited land; therefore, I plant both maize and a few vegetables on my farmland to maximize my har-vest”. 

## 4. Discussion

Africa is identified as a region highly vulnerable to climate change. Floods, erosion, and droughts are becoming more prevalent and threaten the entire country of Nigeria, resulting in increasingly severe consequences [8] for agriculture production. Small-scale agriculture is crucial to the well-being of the people of Nigeria. In a place where food security has already become an issue, climate change has the potential to make this much worse. There is a scientific consensus that the global climate is changing and expected to have substantial impacts on food crop production significantly but in uncertain ways. The trend analysis from this study shows that climatic variables are fluctuating, and this fluctuation is currently affecting and may continue to impact crop yield.

The final selected linear regression models included minimum temperature for both cassava and maize yield. The model for maize additionally included rainfall as a predictor variable. One study indicated that maize yield reduces under heat stress as it impacts both pollination and seed germination [39]. Another study showed that a 1 °C increase in average temperature resulted in a ten percent loss in maize yield [40]. Unlike maize, the model for cassava did not include rainfall as a predictor variable. Cassava is a drought-tolerant species that thrives favorably in harsh climatic conditions [41,42]. This may be due to the crop’s ability to develop large underground roots and its ability to delay leaf production until the next rainfall [43]. It be noted that while cassava is drought tolerant, different phenotypes respond differently to water stress [44]. Chikezie et al. [8] carried out a similar study to this one but in Calabar, Nigeria and recorded similar results on food crop production.

The farmers who took the survey tended to express more concern about irrigation for vegetables rather than either cassava or maize. They reported that while cassava and maize require water at the early stages of development to be successful, they do not require periodic watering, unlike other leafy vegetables. In addition, they showed concern about irregularities in the late onset of rainfall in terms of both frequency and intensity, as this is the factor that defines the planting season. Because of this, the farmers indicated that they often use irrigation at this early stage. Many of the survey respondents expressed concern regarding the danger posed by the increasing incidence of erosion and flooding and how these events have washed away their standing vegetable crops and topsoil in the past, resulting in a decrease in soil fertility. To repair soil fertility, the respondents stated that they have incurred additional costs to apply manure and fertilizer.

Increased atmospheric CO_2_ and climate change may also affect crops indirectly through the impact of pests and diseases [21]. Pests attack food crops leading to reduced yield and a high price of the scarce food crops and consequent malnutrition [34]. Pests were identified as a major concern by the survey respondents, who indicated that they had seen a reduction in the quality and quantity of their crops due to pest infestations. Studies show that there has been an increasing loss in crop yields due to insect pests in a warming climate [45]. As much as a 20–40 percent annual loss of major grain crops such as maize, rice as well as vegetables has been observed in Nigeria [46,47].

On a separate note, the demographics of the survey indicate a decline in farming with younger generations. Less than seven percent of the respondents from this survey were under the age of thirty, yet Waheed [48] estimates the youth unemployment rate to be about fifty percent in Nigeria. The survey respondents discussed their concern over fewer young people entering the profession of farming, yet some discussed situations where youth took over farming from their aged parents, especially in the Badagry community. Wole-Ayo et al. [49] found that in Nigeria, one of the barriers to youth entering agriculture as a profession is the lack of governmental loans. In addition, just under 1 in 4 survey respondents identified as women. This is in line with the results of a study by Anosike and Mayowa [50] which found that 72% of farmers in Lagos are men. Future implications of this aging demographic include a loss of institutional knowledge, but also perhaps an opportunity for creating future opportunities for women to join the profession.

## 5. Conclusions

Our results are limited by the lack of accessible and consistent food production and climate or weather data. Some farmers surveyed were eager to share their almanacs, indicating that they rely on localized information collected by themselves or those they know to make farming decisions. Enete et al. [51] reported that most governments in Nigeria already have agencies charged with environmental issues, including climate change and they most often deliver information through radio and television programs. While up to thirty percent of those surveyed indicated that they had heard of climate change through television or radio, other respondents stated that they pay little attention to these communication formats due to additional expenses incurred in implementing some of the solutions from the information. A lack of consistent data and access to timely information makes it more challenging for farmers to adapt.

## Figures and Tables

**Figure 1 foods-11-03987-f001:**
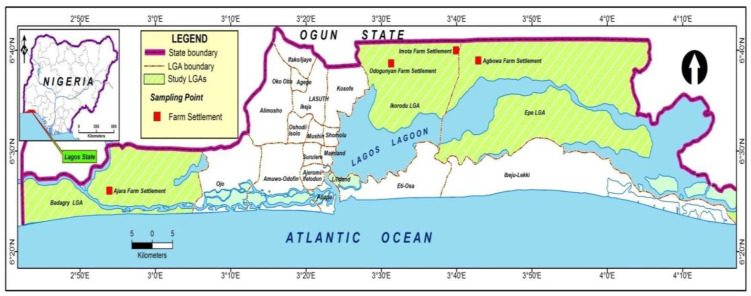
The map of the study area. Source: Author’s Fieldwork (2019).

**Figure 2 foods-11-03987-f002:**
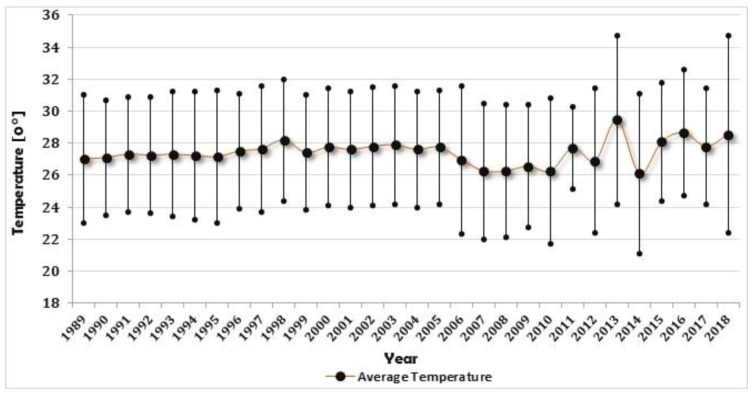
Average annual and corresponding minimum and maximum temperature for Lagos (1989 to 2018).

**Figure 3 foods-11-03987-f003:**
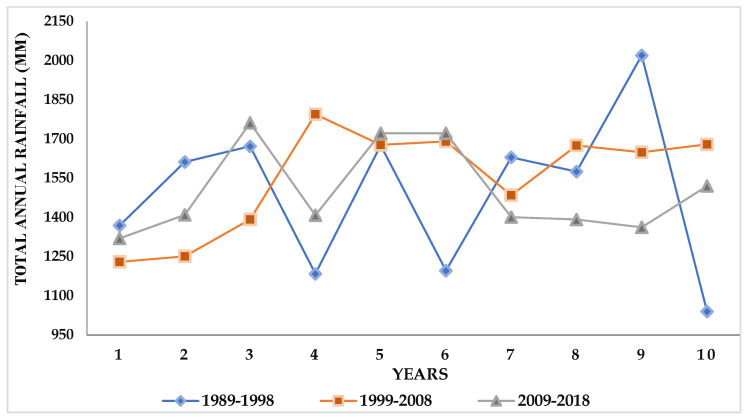
Total annual rainfall for Lagos (1989 to 2018).

**Figure 4 foods-11-03987-f004:**
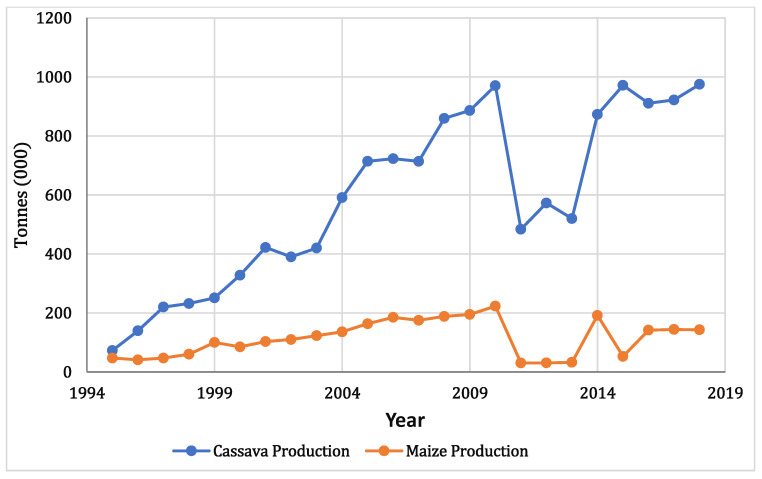
Lagos Cassava and Maize production through time (1995–2018).

**Figure 5 foods-11-03987-f005:**
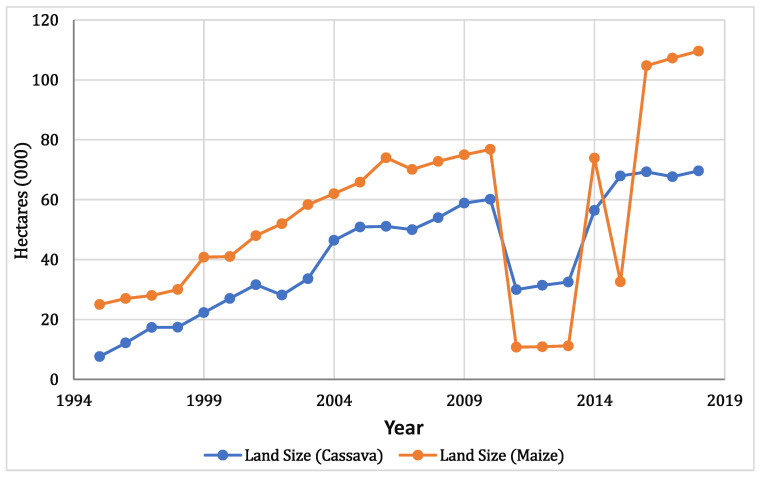
Lagos Cultivated land area for Cassava and Maize production through time (1995–2018).

**Figure 6 foods-11-03987-f006:**
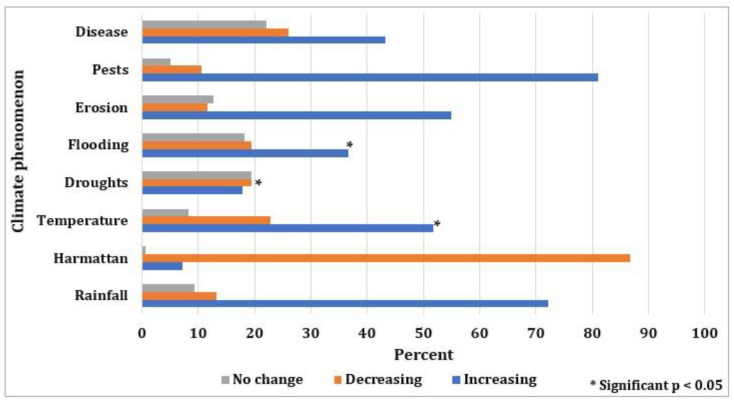
Perception of farmers on the impacts of climate change. Where (*) represents the signifi-cance of the variables at a 95% confidence level, and years of farming experience is used as the determining factor for variation in perception/response.

**Figure 7 foods-11-03987-f007:**
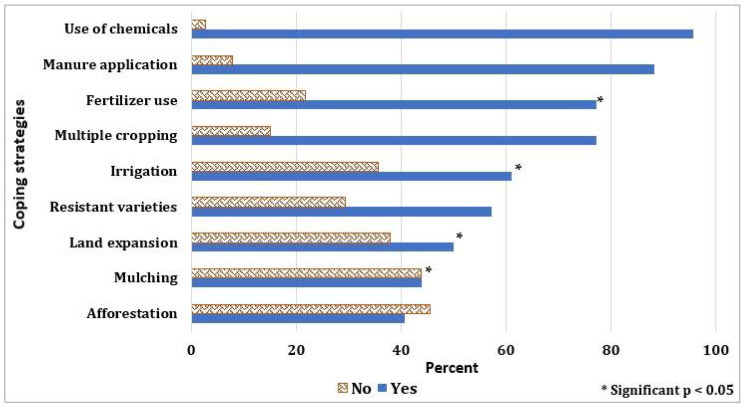
The various identified coping strategies adopted by farmers. Where (*) represents the significance of the variables at a 95% confidence level, and the type of crop grown is used as the determining factor for variation in adoption/response.

**Table 1 foods-11-03987-t001:** Adjusted R^2^ values for all possible linear regression models for cassava production.

Model	Adjusted R^2^	*p*-Value
C = b_0_ + b_1_T_min_	0.1084	0.0642
C = b_0_ + b_1_T_min_ + b_2_A_c_	0.9647	<0.0001
C = b_0_ + b_1_T_max_	−0.0355	0.6495
C = b_0_ + b_1_T_max_ + b_2_A_c_	0.9525	<0.0001
C = b_0_ + b_1_R	−0.0372	0.6802
C = b_0_ + b_1_R + b_2_A_c_	0.9443	<0.0001
C = b_0_ + b_1_T_max_ + b_2_R	−0.0759	0.8295
C = b_0_ + b_1_T_max_ + b_2_R + b_3_A_c_	0.9535	<0.0001
C = b_0_ + b_1_T_min_ + b_2_R	0.0886	0.1451
C = b_0_ + b_1_T_min_ + b_2_T_max_	0.0939	0.1366
C = b_0_ + b_1_T_min_ + b_2_R + b_3_A_c_	0.9638	<0.0001
C = b_0_ + b_1_T_min_ + b_2_T_max_ + b_3_R	0.0744	0.2171
C = b_0_ + b_1_T_min_ + b_2_T_max_ + b_3_R + b_4_A_c_	0.9617	<0.0001

**Table 2 foods-11-03987-t002:** Adjusted R^2^ values for all possible linear regression models for maize production.

Model	Adjusted R^2^	*p*-Value
M = b_0_ + b_1_T_min_	0.3450	0.0020
M = b_0_ + b_1_T_min_ + b_2_A_m_	0.7696	<0.0001
M = b_0_ + b_1_T_max_	0.0028	0.3136
M = b_0_ + b_1_T_max_ + b_2_A_m_	0.7514	0.0000
M = b_0_ + b_1_R	−0.0450	0.9254
M = b_0_ + b_1_R + b_2_A_c_	0.6557	<0.0001
M = b_0_ + b_1_T_max_ + b_2_R	−0.0442	0.6056
M = b_0_ + b_1_T_max_ + b_2_R + b_3_A_m_	0.7506	<0.0001
M = b_0_ + b_1_T_min_ + b_2_R	0.3186	0.0070
M = b_0_ + b_1_T_min_ + b_2_T_max_	0.3297	0.0058
M = b_0_ + b_1_T_min_ + b_2_R + b_3_A_m_	0.7581	<0.0001
M = b_0_ + b_1_T_min_ + b_2_T_max_ + b_3_R	0.3006	0.0170
M = b_0_ + b_1_T_min_ + b_2_T_max_ + b_3_R + b_4_A_m_	0.8223	<0.0001

**Table 3 foods-11-03987-t003:** Demographics for the survey respondents.

	Attributes	Frequency	Percent
Gender	Female	43	23.9
	Male	137	76.1
Age	21–30	12	6.7
	31–40	38	21.1
	41–50	50	27.8
	51 and above	80	44.4

**Table 4 foods-11-03987-t004:** Income class for the survey respondents.

Location	Income (N=)	Frequency	Percent
Epe	<20,000	16	26.7
	21,000–500,000	31	51.7
	51,000–80,000	11	18.3
	81,000–100,000	2	3.3
Ikorodu	<20,000	10	15.0
	21,000–500,000	35	23.3
	51,000–80,000	12	21.7
	81,000–100,000	3	40.0
Badagry	<20,000	29	48.3
	21,000–500,000	20	33.3
	51,000–80,000	9	15.0
	81,000–100,000	2	3.3

**Table 5 foods-11-03987-t005:** Education, experience, and awareness metrics.

Factors	Attributes	Frequency	Percent
Level of education	No Formal Education Primary	17 24	9.4 13.3
	Secondary Tertiary	56 83	31.1 46.1
Years of farming experience	1–5	20	11.1
	6–10	15	8.3
	11–15	12	6.7
	16 and above	133	73.9
Climate awareness	No	35	19.4
	Yes	143	79.4
	Not sure	2	1.2

## Data Availability

The supported data for the reported results can be found at (a) National Bureau of Statistical [52,53], (b) Central Bank of Nigeria (CBN) Statistical Bulletin (2016) [54], (c) Lagos Bureau of Statistics [55].

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
