# Peer review of "The Effect of Climate Change on Food Crop Production in Lagos State"

_foods, 2022, doi:10.3390/foods11243987_

Round 1
Reviewer 1 Report
I commend the authors Tajudeen et al. of the manuscript titled “The effect of Climate Change on Food Crop Production in Lagos state” for their interesting analyses and data gathering about the impact of climate change factors on the production of cassava and maize in Nigeria
The study revealed no clear association between the climate change factors and significant reduction to the production of cassava and maize crops in Lagos. However, there were steady increases in the production of these crops due to several factors including the use of chemical fertilizers. The production of these crops is still dependence on rainfall and developing novel tools for the reducing the dependence on rain fall are essential.
The manuscript is will written in general.
In the introduction, the topic is hot, but the novelty of the work need to be highlighted more compared to previous studies in Nigeria and Africa in general.
In the materials and methods part:
I feel that other factors should be included that may influence the production of such essential crops for example, the role of the ministry of agriculture in Nigeria including (agriculture extension, financial support, hybrid seed production, development of new cultivars.
The survey questionnaire should be supplied as supplementary material.
The conclusion of the work still large and need to be reduced.
-
I give you major revision
Author Response
Point 1: In the introduction, the topic is hot, but the novelty of the work need to be highlighted more compared to previous studies in Nigeria and Africa in general.
Response 1: Additional information about the contribution of the current study to existing research has been added to the introduction section.
Point 2: In the materials and methods part:I feel that other factors should be included that may influence the production of such essential crops for example, the role of the ministry of agriculture in Nigeria including (agriculture extension, financial support, hybrid seed production, development of new cultivars.
Response 2: Some of these factors that could have been explored in the study don’t have annual data that could be used to support the analysis. There is no consistent time series data to back this type of discussion up. However, since yearly crop production depends highly on the total crop area, land size for each crop was used as the control variable. A new multiple linear regression model was fitted for both variables using the climate and cultivated land area data. Also, the result from the survey was used to back up the discussion.
Point 3: The survey questionnaire should be supplied as supplementary material.
Response 3: Please provide your response for Point 2. (in red): The GitHub link to the survey document has been added to the supplementary material section.
Point 4: The conclusion of the work still large and need to be reduced.
Response 4: The conclusion has been reduced from 402 words to 140 words.

Reviewer 2 Report
This paper tries to estimate the effect of climate change on food crop production. The data is fine. However, there are a few questions need to be addressed.
First, the authors do not state clearly their contributions to the literature. This should be added either in the introduction section or in the literature review section.
Second, the biggest issue is that the authors only use a multiple linear regression model that does not include any control variables to estimate the effect of climate change on food crop production. This can only reflect the correlation between climate change and food crop production, not the causal relationship between climate change and food crop production. The authors should add some control variables and rerun the regression.
Third, Table 4 is not the main concern of the paper and it should be deleted.
Author Response
Point 1: First, the authors do not state clearly their contributions to the literature. This should be added either in the introduction section or in the literature review section.
Response 1: The contribution has been added to the introduction section and it goes thus: This study builds on the body of existing research in Nigeria and contributes additional information for Lagos (an area not previously studied) that can be used to inform policy- and decision-makers about climate change-induced agricultural productivity loss.
Point 2: Second, the biggest issue is that the authors only use a multiple linear regression model that does not include any control variables to estimate the effect of climate change on food crop production. This can only reflect the correlation between climate change and food crop production, not the causal relationship between climate change and food crop production. The authors should add some control variables and rerun the regression.
Response 2: Considering that the annual crop production depends highly on the total crop area, land size for each crop was used as the control variable. Some of the points that capture this comment include, “There was a consistent rise in the cultivated area over time for both Cassava and Maize from 1995 to 2011 and a decline from 2013. The average land size for cassava production is 42 thousand hectares with a standard deviation of 19, while the average land size for maize production is 55 thousand hectares with a standard deviation of 30 (Figure 5). The land area for cassava was 7.654 thousand hectares in 1995, and it increased gradually to 69.67 thousand hectares in 2017. Also, the land area for maize production increased gradually from 25 thousand hectares in 1995 to 109.62 in 2018. This trend also corresponds to the production output recorded for both crops”.
Additionally, new linear regression models were fit for each crop and discussed as necessary in the method, result, and discussion sections. Though some other climate variables would have also been explored, the study is limited to temperature and rainfall due to limited access to other data for this study area.
Point 3: Third, Table 4 is not the main concern of the paper and it should be deleted.
Response 3: Table 4, which is now table five, represents the demographic characteristics of the farmer. It was explored to understand how their socio-demographic characteristic could influence their perception about the subject of study. It was not initially cited in the text, making it seem irrelevant. The needful has been done, and it has been mentioned in the text appropriately.
Additionally, one of the authors is a native English speaker. Hence, extensive editing of the English language and style has been done.

Round 2
Reviewer 1 Report
Accepted for me
Reviewer 2 Report
I am fine with the responses.